# The Practice of Physical Exercise and Sports in Portuguese Trans Youth: A Case Study

**DOI:** 10.3390/healthcare11050668

**Published:** 2023-02-24

**Authors:** Joana Oliveira, Roberta Frontini, Miguel Jacinto, Raul Antunes

**Affiliations:** 1ESECS, Polytechnic of Leiria, 2411 Leiria, Portugal; 2CIEQV-Life Quality Research Centre, Polytechnic of Leiria, 2411 Leiria, Portugal; 3Center for Innovative Care and Health Technology (ciTechCare), Polytechnic of Leiria, 2411 Leiria, Portugal; 4Faculty of Sport Sciences and Physical Education, University of Coimbra, 3040 Coimbra, Portugal

**Keywords:** transgender, gender identity, barriers, motives, facilitators, exercise, sport

## Abstract

The following case study aims to analyze the experience of a Portuguese trans individual regarding their practice of physical exercise (PE) and sports in Portuguese gyms and sports clubs. A 30-min interview was conducted through the Zoom platform. Before the interview, four questionnaires—Satisfaction with Life Scale (SWLS), Positive and Negative Affect Schedule (PANAS), Hospital Anxiety and Depression Scale (HADS), and EUROHIS-QOL 8-item index—were also applied, all in their Portuguese version. The interview was digitally video recorded after consent was obtained, transcribed verbatim, and subject to thematic analysis. Findings suggest positive values for satisfaction with life and quality of life. The values of positive affect were higher than those of negative affect, and there was an absence of depressive and anxious symptomatology. In the qualitative analysis, mental health was the main motive for the practice, while locker rooms separated by gender and university life were the main barriers mentioned. Mixed changing rooms were identified as facilitators of PE practice. This study highlights the importance of developing strategies for the creation of mixed changing rooms and sports teams in order to promote a comfortable and safe practice for all individuals.

## 1. Introduction

The trans community represents between 0.1% and 0.2% of the general population [1]. In Portugal, trans individuals have been identified as one of the most discriminated against and excluded populations [2]. In fact, studies report situations of discrimination, violence and transphobia toward trans individuals [3,4,5], which leads this population to avoid particular spaces and public facilities, such as gyms [6,7]. 

The term transgender or trans refers to a person whose gender identity differs from the sex assigned to them at birth [8,9,10]. A trans man is an individual with female birth-assigned sex but identifies as male, and a trans woman is a person with male birth-assigned sex but identifies as female [9].

In the 2015 U.S. Transgender Survey, 14% of respondents reported that they did not go to a gym or health club in the past year because they were afraid of meeting a trans person [11]. Nearly one in five (18%) trans individuals who went to a gym or health club where staff knew or thought they were trans reported being denied equal treatment or service, being verbally harassed or physically attacked because of being trans [11]. In the study by Symons and colleagues [12], almost half of the participants from mainstream clubs (48.2%) reported that their club was unwelcoming to trans individuals, and 58.3% of trans respondents reported that there were sports that they did not play because of their gender identity. In the same study, the two-sexed/gendered sports model was presented several times as difficult for trans people [12]. Participants of the study also highlighted general ignorance and prejudice concerning trans issues within many of the sporting communities, experiences of discrimination, a lack of policies to enable their participation in sports, concerns with using changing rooms and concerns regarding being accepted and fitting in [12].

Trans people have more barriers than reasons to engage in PE and sports, the most frequent being body image dissatisfaction [13]. This is not surprising, as trans people tend to be more dissatisfied with body image and have a higher prevalence of eating disorders [14]. In addition, changing rooms are frequently reported as unsafe and uncomfortable spaces that limit the regular practice of PE and sports among trans individuals [13,15,16,17]. In 2017, in the British school context, it was reported that 29% of trans students are bullied during sports lessons, especially in changing rooms (25%) [18]. Negative past experiences are also an aspect reported in the literature as one of the main barriers to the practice of physical exercise [12,19], which cause acute feelings of gender dysmorphia, especially in a school context [19]. In elite competitive sports, the debate and discussion about the participation or not of trans athletes have increased, as well as scientific evidence on the possible advantages of trans women in sports competitions [20,21]. Although the policies and regulations for participation in the Olympic Games have changed over the years [22], participation rules still discriminate against and exclude transgender people from sports [12]. On the other hand, despite the immense barriers identified in the literature, it seems that the practice of PE and sports activities is empowering and may play a fundamental role in the gender identification process [23].

Therefore, and considering the aforementioned data, research must address these issues in order to achieve a more inclusive and comfortable practice for all individuals. However, there is still a long way to go, since this is a field that has been that has been neglected for many years. A recent systematic review [13] included articles mainly from the UK and USA, with results related to their population. To the best of our knowledge, there is no evidence of studies on this topic in Portugal. With the purpose of knowing the reality of the Portuguese context and helping make it more inclusive and safer for the trans population, the present study aims to shed light and analyze the experience of a Portuguese trans individual regarding his practice of PE and sports in Portuguese gyms and sports clubs, identifying possible barriers, motives and facilitators for the practice.

## 2. Materials and Methods

The present study was conducted in accordance with the Declaration of Helsinki and its subsequent amendments [24]. In order to report qualitative data, the COREQ (Consolidated Criteria for Reporting Qualitative research) checklist was followed [25].

### 2.1. Study Design and Participant

This was a qualitative exploratory case study conducted with a trans male participant (n = 1) who was involved in the practice of PE and had a history of sports practice. The participant was 17 years old, identified as transgender, and was a practitioner of physical activity (PA) and/or PE and/or sport. The participant was recruited via advertisements on social media. After expressing interest, the participant contacted the authors for a more detailed explanation of the study.

The participant met all criteria and agreed to participate. A meeting was scheduled through the Zoom platform and was video recorded. Written informed consent forms were obtained before any data collection took place.

### 2.2. Data Collection

Data were collected in June 2022. Before the interview, four questionnaires were also completed—the Portuguese version of the Satisfaction with Life Scale [26]; the short Portuguese version of the Positive and Negative Affect Schedule (PANAS) [27]; the Portuguese version [28] of the Hospital Anxiety and Depression Scale; and the Portuguese version of the EUROHIS-QOL 8-item index [29]—to measure global life satisfaction, both positive and negative affect, anxiety and depressive symptoms and quality of life, respectively.

Information on gender, age, marital status, physical activity (PA), physical exercise (PE) and sports practiced was collected. For the purpose of this study, the participant was asked about five domains: the frequency and type of PA, PE and sports practiced before and after gender affirmation; barriers to its practice; facilitators of its practice; motives for its practice; and safety of the practice spaces. The participant was free to respond in ways he felt comfortable with and was encouraged to present his own understandings and meanings. The researcher could explore a particular meaning or sentence and could clarify any question raised. The interview was conducted by one health and clinical psychologist and psychological field (RF) with qualifications in qualitative research, experience in sport and exercise psychology research and interviewing. To reduce any influence on the participant’s responses, the researcher had no previous contact with the participant, and only the participant and the researcher were present during the interview. A semi-structured guide was used to explore the participant’s perspectives. Specialists reviewed this interview in both areas of psychology and sports. The session lasted 30 min. A digital video and audio recording of the interview was made with the participant’s permission, for later transcription and analysis.

### 2.3. Data Analysis

The interview was transcribed verbatim and analyzed using thematic analysis [30]. Two authors independently performed data analysis. However, the entire research team organized and conducted regular online meetings to examine and discuss topics regarding data analysis to ensure reflexivity. Data analysis occurred in three phases: data reduction, data display, and conclusion drawing/verification [30]. In the first phase (i.e., data reduction), significant sections of the answers were coded into themes using a deductive–inductive approach. All responses were meticulously read in order to obtain a complete picture of the participant. Furthermore, interpretative notes were made. Subsequently, emerging subthemes were grouped into major themes. The data display permitted drawing conclusions. The map’s themes and quotations were created to organize and facilitate data analysis. In “conclusion drawing/verification” researchers reviewed the meaning and significance of the data analyzed. Emergent conclusions were confirmed as a means of testing the validity of the findings [30]. Both themes and subthemes were examined, reviewed and iterated to guarantee that they reflected the data collected. After independent analysis, researchers compared and discussed the findings to reach a consensus.

## 3. Results

At the time of the study, the participant was a young trans male who was regularly engaged in PA and PE practice. On average, the participant practiced PE for five years, three times/week and four hours/week. Regarding the practice of PA, the participant practiced physical activities, such as running and cycling, three hours/per week.

Table 1 summarizes the descriptive statistics related to subjective well-being values (life satisfaction, positive and negative affect), anxiety and depression symptoms and quality of life.

Regarding satisfaction with life, the average was 4.60, while regarding the emotional dimension of subjective well-being, the values of positive affect were higher than those of negative affect. Regarding anxiety and depressive symptomatology, anxiety symptoms (6.00) had a higher score than depressive symptoms (3.00). Finally, the quality of life mean value was 81.25 (on a scale ranging from 0 to 100).

Thematic analysis revealed five main themes: practice of PE and sports, barriers to engaging in PA and sports, facilitators of PE, motives for PE practice and safety of the places. Nineteen subthemes were also identified. The results, organized by themes, subthemes, illustrative expressions of the interview, and the number of times the theme appeared, can be found in Table 2.

### 3.1. The Practice of Physical Exercise and Sports

In this topic, subthemes emerged before gender affirmation (N = 1) and after gender affirmation (N = 1). Before gender affirmation, the participant practiced sports (specifically karate and futsal) and PE at a gym. After gender affirmation, the participant abandoned sports and remained only in the gym.

### 3.2. Barriers to Engaging in Physical Activity and Sports

Regarding barriers, six subthemes were identified: locker rooms divided by gender (N = 7), university life (N = 4), COVID-19 pandemic (N = 1), PT behavior (N = 1), sports characteristics (N = 1) and gender division in sport (N = 1). The most discussed subtopic throughout the interview was locker rooms, followed by university life. The participant feels uncomfortable taking a bath in the changing rooms. In addition, the participant presents the irregular college schedulers as a factor that hinders a more assiduous practice of PE.

### 3.3. Facilitators of Physical Exercise

Regarding facilitators of PE, five subthemes were identified: mixed changing rooms (N = 3), gym environment (N = 2), city environment (N = 1), social experiences (N = 1) and mastectomy (N = 1).

### 3.4. Motives for Physical Exercise

Five subthemes related to motives for PE practice were identified: mental health (N = 3), energy release (N = 2), health (N = 1), desire to achieve a specific physical form (N = 1) and pleasure (N = 1). The participant reported mental health maintenance as the main motive for exercise practice.

### 3.5. Safety of the Places

Regarding the safety of the places, one subtheme was identified: gyms as a safe place (N = 1).

## 4. Discussion

The present study case aimed to fill a gap in the literature regarding knowledge of the experience of the practice of PA, PE and sports in trans individuals. This study aimed to help create more inclusive and safer spaces for the practice of PE and sports, explicitly considering the higher levels of sedentary behaviors and the lower practice of PE and sports of the entire population. Thus, some barriers were identified in the present study, which can also be found in a recent systematic review of the topic by Oliveira et al. [13].

### 4.1. Satisfaction with Life

Regarding satisfaction with life, the average was 4.60, which means that the individual was slightly satisfied with life. However, the literature indicates that trans individuals tend to report reduced levels of satisfaction with life [31], which is easily explained since trans individuals tend to experience situations of discrimination [5], abuse and violence [4], and difficulties in accessing health and employment opportunities [3]. However, the slight satisfaction reported by the participant in the present study can be explained by the results of the study by Drydakis [32], which suggest that changing one’s appearance to match gender identity and transitioning can be positively associated with life satisfaction. In addition, parental support regarding gender identity and expression seems to be related to greater life satisfaction in trans individuals [33].

### 4.2. Well-Being and Quality of Life

In the emotional dimension of subjective well-being, the values of positive affect were higher than those of negative affect, and there was an absence of depressive and anxious symptomatology. The participant seemed to be slightly satisfied with life, and this topic was positively related to positive affect [34] and negatively correlated with depression and anxiety [35]; in the same sense, these results were also not in accordance with the literature, since this population presents high levels of mental health problems [3,36,37], namely, with the presence of anxious and depressive symptomatology [38]. With regard to the quality of life, despite the participant in the present study having a good quality of life, evidence suggests that transgender people have a lower quality of life than the general population [39]. Hormone therapy has been associated with increased quality of life and decreased depression and anxiety [40,41], which may explain all the positive values that the participant presented related to these topics, since hormone treatment had already started.

### 4.3. Type of PA Practiced

The differences regarding this aspect before and after gender affirmation seemed to play an essential role in characterizing transgender individuals’ participation in PE and sports [42]. Individual activities, such as the gym, and collective and individual sports, such as futsal, karate, and dance, were the activities that the participant of this study performed until his gender affirmation. After that, sports were no longer part of his PE routine. In a study by López-Cañada and colleagues [42], their results revealed that the participation of trans people in unorganized PA and sports activities is higher than in organized activities after gender disclosure. In the same study, it was also reported that the activities most practiced after gender disclosure were jogging, walking and bodybuilding [42]. These results can be justified by the fact that trans individuals tend to avoid social interaction [17] due to fear and anxiety about the reaction of other people [43,44,45].

### 4.4. Barriers to Practice

The results of this study were in accordance with the recent systematic review by Oliveira and colleagues [13], especially regarding external barriers. Inadequate changing facilities and regulations surrounding gender were some of the obstacles mentioned in the present study, corroborating the findings of the previous systematic review. The bathhouse continued to be the most mentioned subject, with the structure and gender division of the changing rooms mentioned seven times by the participant and presented as the barrier that generates the most concern. This is supported by previous literature [12,13,16,43,46,47] and, consequently, the aspect that should have most attention in the context of practice. Changing rooms and bathrooms, characterized as awkward or unsafe places that produce a fear of social rejection [17,42], and where many trans people experience embarrassment and stress [15], are sites of hateful violence against trans and non-conforming people [48]. Furthermore, the binary system and the lack of private curtains are factors that create inequality and exclusion of trans children in the school context [49]. In order to solve this problem, *“there could even be mixed changing rooms”* as suggested by our participant. Neutral or mixed changing rooms with structures that provide greater privacy are the PE facilitator most mentioned in this study, and also a strategy presented several times in other articles (e.g., [16,17,47]. To make changing rooms more comfortable and safer for everyone, strategies, such as those mentioned above, should be considered and implemented in different contexts of PE and sports by fitness organizations and sports clubs, or even in educational contexts.

The binary structure is not only a characteristic of changing rooms but also of the sports context and culture [19]. The gender division in sports is based on the assumption that sex and gender are binary and immutable characteristics and that physical differences between men and women substantially affect athletic performance [50]. In this study, it was reported as a barrier to trans individuals’ participation in sports, as it limits participation to people whose sex does not match their gender identity. This is an important aspect for sports organizations to consider. Mixed sports should be a strategy for fair and inclusive participation.

Some sports, notably swimming (as confirmed by the participant in the present study), are less popular among transgender individuals due to excessive body exposure [51], and swimming pools are often considered unsafe spaces in both pre- and post-transition phases [15]. Therefore, it is important that sports organizations create inclusive strategies and spaces where everyone feels comfortable during practice. *“The mastectomy will open me many doors”.* In this study, performing upper surgery is presented as one of the facilitators of PE practice. This is a factor also presented in the study by Jones and colleagues [44], where two participants who had chest reconstructive surgery reported feeling more comfortable and engaged in more PA. This is important data for healthcare systems in order to provide conditions and quick access to gender affirmation programs. A qualitative study by Elling-Machartzki [15] also demonstrated the importance that the transition phase has in PA and sports engagement. In fact, participants felt more comfortable with their bodies during PE and sports after the transition [15]. These data are extremely important, especially for healthcare systems. New inclusive policies and conditions should be considered to facilitate the process of gender-affirming surgery and care, as it seems to influence the quality of life of trans individuals.

University life was another barrier mentioned several times (n = 4) in this study. The transition from high school to university, lack of time, and irregular schedules characteristic of college life seem to be the main reasons that prevented more active participation [52,53,54]. In addition, the COVID-19 pandemic was also presented as one of the factors limiting practice, which is not surprising, since it negatively impacted PA [55]. This is supported by a recent systematic review [56], where the results report a decrease in PA and an increase in sedentary behaviors of several populations during confinement.

In the present study, the worst experience mentioned by the participant in the context of PE was related to the behavior of a personal trainer, which presented itself as another barrier to practice, and this professional may determine their continuity or drop out [57]. This is important, especially when dealing with a sexual minority, such as trans people, who mostly have negative experiences in the context of sports and PE [13,19,58]. On the other hand, social experiences, in addition to being a facilitator of PE identified in this study, were also the participant’s best experiences. In a study by Unger and Johnson [59], the results suggested that friendships that involve exercising together and the social contacts that result from the practice of PE can motivate exercise behavior.

### 4.5. Reasons to Practice

The practice of PE has numerous benefits regarding the maintenance or improvement of mental health [60], and this was the most discussed theme throughout this study. In the literature, regular practice of PE and PA is associated with improved mental health and the prevention of anxiety and depression disorders [61,62,63]. In addition to this topic, physical health was also mentioned as one of the reasons to practice. The importance of practicing PA and PE in order to prevent several diseases, such as cardiovascular diseases, diabetes and obesity, was evident [64,65,66]. The fact that physical and mental health were the main reasons for practicing PE in this study suggests that people, especially trans people, are increasingly recognizing the key role that PA and PE can play in maintaining and improving mental and physical health. This awareness is extremely important to promote more active and healthier lifestyles in the future.

In the present study, pleasure was identified as one of the reasons for the practice: *“Because I like it”*. Although the trans population does not refer to it as one of the main factors that promote the practice [47], the link between enjoyment and adherence and continuity of practice in different contexts of PA has been studied over time. Studies suggest that enjoyment seems to be a relevant predictor for the continuity and adherence to PA in gyms [57,67] and in the sports context [68]. Moreover, the desire to achieve a specific physical form, which was another reason for the practice identified in this study, is one of the most mentioned topics in the literature [15,45,47,69]. In this case, similar to those studies, the aesthetic issue is not only related to a construction of their body image with a social stereotype, but with the need to accelerate the changes produced by hormonal treatment.

### 4.6. Facilitators to Practice

The gym and city environment were also identified as facilitators of practice. The fact that there is no sex segregation in the gym makes it a more neutral space compared to sports that are divided between men and women. This finding in the present study may explain the results of the study by López-Cañada and colleagues [42,46], who reported a decrease in the participation of transgender individuals in sports such as football and basketball, and an increase in bodybuilding after gender disclosure. In addition, geographic factors seem to influence the practice of PE and sports. The results of this study suggest that cities with a greater diversity of people end up becoming safer and more comfortable spaces for trans people compared to cities with higher rates of transphobia and lower diversity. In fact, the environment in which the exercise takes place seems to influence the practice experience and the practitioner’s behavior [57], especially in trans individuals who frequently experience situations of discrimination, abuse and violence [3,4].

In the present study, there were no reports of harassment, discrimination or transphobia. According to the participant of this study, Portuguese gyms and sports clubs are usually safe places where *“people don’t want to know who you are”*. Although the experience reported in the present study was mostly good, there are still aspects, which are also reported in the existing literature, that require the attention of the sports and fitness community, especially regarding sports and PE facilities. The binary system of changing rooms and the lack of privacy in them is something that seems to concern trans individuals, as well as the sex segregation existing in sports. It is necessary to find strategies that correspond to the needs of practitioners and that guarantee safety, comfort and inclusion for all, considering the creation of mixed changing rooms with private cubicles for bathing and changing, and the creation of mixed teams.

## 5. Practical Implications

Case studies are important for research, since they capture a wider range of perspectives, contrary to the use of surveys. Therefore, this study may have important implications for the practice. Firstly, it may be important for gyms and sports clubs to have a greater sensitivity to these issues. They may minimize some behaviors so that trans individuals can feel safer during the practice of sports. For example, it may be important to consider creating specific spaces for non-binary people (e.g., gender-neutral changing rooms). It is also crucial that both gyms and sports clubs assess these perceptions of their members very carefully in order to enhance the benefits of sports and physical exercise. Namely, the importance of focusing on the benefits for mental health is highlighted, since this was an emerging point from this study. On the other hand, this study draws attention to possible threats and barriers to the practice of sports that must be addressed. The presence of friendlier spaces for showering and changing clothes and the decrease of gender division in team sports (or the creation of mixed teams) should be strategies to be considered. In addition, improving inappropriate behaviour by PTs is important and should be a priority. In an increasingly inclusive world, where barriers are being broken down, and it is becoming easier to assume one’s gender identity, it is possible that the number of difficulties for trans individuals to participate in PE and sporting activities will increase. Thus, it is of the utmost importance to better understand the specific challenges that a transgender individual may experience when engaging in sports activities and PE. This study draws attention to research questions that should be further explored. It is important that future studies look into these issues, so that exercise and sports are safe practices for all individuals and enjoyable moments. This way, their benefits (not only physical but also psychological) will be enhanced.

## 6. Limitations and Future Research

Despite its contributions to research and practical implications, this study has some limitations that should be acknowledged, and that can guide future research. Firstly, as a case study, the present study only analyzed the experiences of one trans individual and cannot be generalized, which means that the results should not be generalized to the entire trans community. Thus, it is important that future research includes larger samples in various contexts. Reaching out to LGBTQ+ associations or other kind of organizations where these issues are important could be a way to achieve a more representative sample. Secondly, this study only provided the perspective of trans men in its specific context. In future research, it is important to identify the barriers, motives and facilitators that trans women may also be facing, as well as the differences between the experiences of trans women and trans men in sports and exercise contexts. Finally, the inclusion of people surrounding the trans individuals (e.g., coaches, fitness and sports managers, physical education teachers, etc.) should be considered in order to have the perspectives of sports and PE professionals and a better understanding of the existing sports policies in these contexts.

## 7. Conclusions

After gender affirmation, there are changes in the amount and type of physical activities practiced by trans individuals. Changing rooms divided by gender and university life may be factors that hinder a comfortable and regular practice of PE and sports. The characteristics of the sports, the behavior of personal trainers, and the COVID-19 pandemic were also identified as possible barriers to practice. Furthermore, mental health was the most frequently mentioned reason to practice PE throughout this study. Besides this, it seems that the creation of mixed changing rooms and mixed sports teams would be an inclusive strategy that would facilitate the practice of PE for transgender people.

## Figures and Tables

**Table 1 healthcare-11-00668-t001:** Summary of the descriptive statistics.

Hours of weekly PA practice	3.00
Years of PE practice	5.00
Number of weekly PE practice	3.00
Hours of weekly PE practice	4.00
Satisfaction with life	4.60
Positive affect	2.80
Negative affect	1.40
Anxiety symptoms	6.00
Depression symptoms	3.00
Quality of life	81.25

**Table 2 healthcare-11-00668-t002:** Qualitative analysis of the interview.

Main Theme	Subthemes	Illustrative Expression	N
Practice of physical exercise and sports	Previous	*“I started to practice sports and physical exercise when I was 14/15 years old. (…) Sports were karate and futsal. And physical exercise, gym. (…) And dance, yes.”*	1
	After gender affirmation	*“I don’t do any sports anymore; I just go to the gym. (…) I usually go 2/3/4 times. I want to go; I will go. Other times, “I have to go on this day to have regularity. (…) I stay as long as I can. I always try to stay 2 h, but sometimes it’s only for 45 min, 1 h and a half… I do more weight training and cardio and occasionally do cycling or something else.”*	1
Barriers	Locker rooms divided by gender	*I haven’t had the courage to take a shower yet because I still don’t have a mastectomy. After the mastectomy, I don’t mind showering there, but other than that, there’s nothing stopping me from being there and doing my thing.”*	7
	University life	*“It’s about the college. I had a very irregular schedule. Now, as I will be in exams, it will be easier to be regular, at more certain times, but I had a very irregular schedule in college, so that was a barrier.”*	4
	COVID-19 pandemic	*“Yes, I live in a village, and here we have a gym, and we signed up for that gym, but because of the pandemic, it was impossible to continue.”*	1
	PT behaviour	*“The worst was perhaps when I went to do my first physical evaluation in the gym. I was a little overweight, and an instructor treated me a bit… I don’t know if it was because I was trans, or I don’t know if he noticed, but I felt a little lack of sensitivity from him… (…) he called me obese, and I felt a little bit sick. When I started the treatment, I started to eat a lot more and with the pandemic, it made it a little difficult to lose weight or maintain weight. I missed his sensitivity a bit, but other than that I haven’t had any negative experiences (…). He was very insensitive. I didn’t like him at all.”*	1
	Sports characteristics	*“I didn’t want to go to the gym, I wanted to do swimming, but obviously, it was a bit more impossible to do this sport for obvious reasons, but of course, yes, of course, there are barriers.”*	1
	Gender division in the sport	*“For example, when there are places that have a football team that until a certain age is all together, boys and girls, and then from that age onwards you start to divide, boys on one side, girls on the other. There is no sport that is totally mixed, or if there is, I don’t know which one, but it is always divided. For example, we are talking more about trans people who identify with their own gender, because that’s my perspective, but if I were a non-binary person, I would come here and say, “I don’t feel comfortable playing football because there is no team that we are all…that is everything”. It’s either a girl or a boy. So yes, it’s still a barrier, even for people like me, who identify as male and I have that in my head, and that’s what I am, but even so, I could want to do football but not want to play in a male team, I might want to play in a mixed team. These issues still exist. We have already seen several examples of trans people who practiced a modality in the Olympic games and who now practice the same modality, but in a male or female team, contrary to what they practiced before, we saw that it does not matter if that person is trans or not because there are cis women who will be “stronger” than cis men, so I feel that the gender issue doesn’t make much sense, because it has already been proven that there will be people of other genders who will be stronger, more agile than you, even if you are of the opposite gender, then I feel like the gender question doesn’t make much sense.”*	1
Facilitators of physical exercise	Mixed changing rooms	*“Portugal is still conservative, I feel that we have made some progress, but even so, people are still very closed to these issues. If there were neutral changing rooms, it would be ok to have a man, a woman, and a non-binary all together in there. It’s a basic need, it’s the bathroom, and it’s for showering. That’s what the locker room is for, it’s not to divide up genders, but obviously, it’s all for the safety of all parties. I don’t mind sharing a locker room with another woman, obviously, but there are people who do, so it might not be safe for us or them.”*	3
	Gym environment	*“And although in this city there are a lot of sports, there are infinite sports there. But I didn’t feel the need to continue in any. As I wanted to maintain my physical shape or wanted to improve, I decided to go to the gym, because I also think it’s more neutral. … in terms of gender.”*	2
	City environment	*“I think it would also depend a lot on where you live, because, for example, I exercise in this city, which is a city where young people from all over the country come to study, and so there ends up being a great flow of diversity of people. I don’t know if there are more trans people in that gym, there probably are, but I’ve never seen them, or I’ve never noticed them, but from my perspective, from what I can say, I feel like it’s safer. Now, for example, a person who is from a more closed city, who has a higher rate of transphobia, I think there may be security issues that can compromise the person.”*	1
	Social Experiences	*“I enjoyed going with my male colleagues. It was very natural. They knew me before I came out of the closet as a trans person, and it was all very natural, and I enjoyed going with them and having that locker room conversation. It was all very natural. I think that was the best experience I had.”*	1
	Mastectomy	*“The mastectomy will open many doors.”*	1
Motives for physical exercise practice	Mental health	*“I exercise to take a moment for myself, to think about my stuff, and I think we’re all like that, no matter the gender, no matter cis or trans. I think everyone will do physical exercise also for a mental issue.”*	3
	Energy release	*“I’ve never liked being quiet, so it’s a good way to spend energy.”*	2
	Health	*“For health too.”*	1
	Desire to achieve a specific physical form	*“As I am already on my hormonal treatment, physical exercise will help me a lot to get what I want, the body I want.”*	1
	Pleasure	*“It’s because I like it.”*	1
Safety of the places	Gyms as safe places	*“I cannot generalize. I’ve only had experiences in two; one was, literally, a garage. I don’t have much to go on. I feel that, in my experience, they are safe, despite the issue of the changing room. In the exercise itself, people don’t want to know who you are.”*	1

Note: N = frequency response in each subtheme.

## Data Availability

Additional data are available to the corresponding author upon request.

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
