# Peer review of "The Practice of Physical Exercise and Sports in Portuguese Trans Youth: A Case Study"

_healthcare, 2023, doi:10.3390/healthcare11050668_

Round 1

Reviewer 1 Report

Given the sensitivity of the topic, ethical approval should be submitted.

Methods There is no data on respondents: number of respondents, age, gender...

The applied psychometric questionnaires were validated in Portuguese, can they be used on such a sensitive sample?

How was the data collected on PA and PE, is it a standard questionnaire or a newly created questionnaire?

Data Analysis It is not necessary to state which author did which part of the research.

Two authors (JO and RF) independently performed data analysis

No method of statistical data processing reported. The tools used allow quantitative analysis and not only qualitative. It would be interesting to see if there is a correlation between the level of PA and satisfaction/dissatisfaction in Life. It would be good for each questionnaire to display descriptive data in a table: median, mode, mode frequency.

The authors claim: No substantial differences were found, with the results being agreed by all researchers. Which test proved that there are no differences?

The data collected by surveys is not displayed as it should be.

The results of the questionnaire can be found in table one, why is only the total score. The table does not indicate whether the total score is shown as a mean. Why is there no other descriptive data.

The table should have the same number of decimal places for all data.

Some frequencies are mentioned under points 3.1. until 3.5. but it is better that these data are in a table. 

The paper contains valuable information that is difficult to obtain, but the method and results chapters should be rewritten.

Author Response

Response to REVIEWER 1

Given the sensitivity of the topic, ethical approval should be submitted.

Response: Thank you for your comment and suggestion. The information regarding the approval of the ethics committee has been added in the manuscript (“The study was conducted in accordance with the Declaration of Helsinki and approved by Ethics Committee of the Polytechnic of Leiria (reference number: CE/IPLEIRIA/20/2022)” ).

Methods There is no data on respondents: number of respondents, age, gender...

Response: Thank you very much for the comment. The information has been added in the Participants section.

The applied psychometric questionnaires were validated in Portuguese, can they be used on such a sensitive sample?

Response: All the instruments used have been translated and validated for the Portuguese population in samples with different characteristics (e.g., young, adult, and elderly; clinical and non-clinical populations), namely:

  • Satisfaction with Life Scale (e.g., Neto, 1993; Antunes et al., 2019);
  • Short Portuguese version of Positive and Negative Affect Schedule (e.g., Galinha et al., 2014; Antunes et al., 2020);
  • Portuguese version of the Hospital Anxiety and Depression Scale (e.g., Pais-Ribeiro, et al., 2007;
  • EUROHIS-QOL 8-item index (Pereira et al., 2011; Antunes et al., 2020; Frontini & Antunes, 2021)

For this reason, and given that they are instruments widely used in different contexts, we consider that these instruments fit the objectives of the present study.

How was the data collected on PA and PE, is it a standard questionnaire or a newly created questionnaire?

Response: Thank you for your comment and the opportunity to clarify this issue. The information regarding the practice of physical activity and physical exercise was collected through an interview.

Data Analysis It is not necessary to state which author did which part of the research.Two authors (JO and RF) independently performed data analysis

Response: Corrected according to the reviewer's suggestions.

No method of statistical data processing reported. The tools used allow quantitative analysis and not only qualitative. It would be interesting to see if there is a correlation between the level of PA and satisfaction/dissatisfaction in Life. It would be good for each questionnaire to display descriptive data in a table: median, mode, mode frequency.

Response: Thank you very much for your comment. Since this is a case study, the idea of using quantitative instruments was only to complete the information that was collected in the interview. Thus, it did not seem appropriate to make any type of analysis other than indicating the values of the domains of each scale. We are available if the reviewer feels that any further information is needed.

The authors claim: No substantial differences were found, with the results being agreed by all researchers. Which test proved that there are no differences?

Response: Thank you very much for your comment. The reviewer is right in his comment. The sentence has been deleted.

The data collected by surveys is not displayed as it should be.

Response: As mentioned in response to previous comments, this is a case study, and the use of quantitative tools was only intended to complement the information gathered. Thus, Table 1 reports the values of each variable (domain) analyzed, with no indication of mean and standard deviation since only one participant was involved.

The results of the questionnaire can be found in table one, why is only the total score. The table does not indicate whether the total score is shown as a mean. Why is there no other descriptive data.

Response: Table 1 presents the data regarding the questionnaires applied to the participant. Given that this is only one participant (case study), the results refer to the individual's total scores in each instrument.

The table should have the same number of decimal places for all data.

 Response: Corrected. Thank you.

Some frequencies are mentioned under points 3.1. until 3.5. but it is better that these data are in a table. 

 Response: Thank you for your comment. All values and frequencies given in 3.1, 3.2, 3.3, 3.4, and 3.5 are given in table 2 (N column). For better understanding, a footnote has been placed on the table.

The paper contains valuable information that is difficult to obtain, but the method and results chapters should be rewritten.

Response: Many thanks for the reviewer's suggestions. We have tried to improve the document according to the reviewer’s suggestions.

Reviewer 2 Report

I enjoyed reading the document, it seemed well written and well structured. However, care must be taken when making definitive conclusions with N = 1. In this sense, review the entire document in order to avoid definitive conclusions/assumptions.

INTRODUCTION:

Please insert one paragraph after the first one defining and describing main characteristics of a trans individual Insert 1-3 references.

Author Response

Response to REVIEWER 2

I enjoyed reading the document, it seemed well written and well structured. However, care must be taken when making definitive conclusions with N = 1. In this sense, review the entire document in order to avoid definitive conclusions/assumptions.

Response:  Thank you very much for your comments. The document has been revised to clarify that this is a case study and not to make generalizations.

INTRODUCTION:

Please insert one paragraph after the first one defining and describing main characteristics of a trans individual Insert 1-3 references.

Response:  Thank you for your comments. The paragraph has been added.

Reviewer 3 Report

This case study is aimed to assess the experience of a Portuguese trans individual regarding their practice of physical exercise and sports in Portuguese gyms and sports clubs. It's not clear that the authors conducted this case study to assess the Portuguese trans experience during practice physical activity or to assess the impact or acceptance of the Portuguese community to accept trans individuals. The aim of the study is not precise and generic. As the authors knew that Portugal approves laws to boost transgender rights in the past 5 years. Therefore, I think being transgender in Portugal is not a crime, they have the law to protect them. I don’t understand the rationale of this case study. However, there are comments and concerns that should be addressed.

The introduction is very long, despite that journal instruction has no limits. The authors reported a lot of information, that makes the readability of this case study very difficult. This introduction could be fit for a full-text article or review. I suggest the authors make it short and focus on the aim of the study.

The same thing in the discussion, the authors reported too much information. This discussion form is not acceptable. The authors should shorten the introduction and discussion section accordingly.

The authors stated that this study is conducted according to the declaration of Helsinki and its subsequent amendments. However, it’s not clear if the author gets the ethical committee approval or not. According to the journal instruction, this type of study should be approved by the IRB.

The outcome measures used in this study should be reported appropriately in the methods section. The authors should report the validity and reliability of these outcomes.

Author Response

Response to REVIEWER 3

This case study is aimed to assess the experience of a Portuguese trans individual regarding their practice of physical exercise and sports in Portuguese gyms and sports clubs. It's not clear that the authors conducted this case study to assess the Portuguese trans experience during practice physical activity or to assess the impact or acceptance of the Portuguese community to accept trans individuals. The aim of the study is not precise and generic. As the authors knew that Portugal approves laws to boost transgender rights in the past 5 years. Therefore, I think being transgender in Portugal is not a crime, they have the law to protect them. I don’t understand the rationale of this case study. However, there are comments and concerns that should be addressed.

Response: Thank you for the opportunity to clarify this point. In fact, being trans in Portugal is not a crime, and there have been legislative changes in recent years, as the reviewer mentions. However, the study by Oliveira et al., 2022 clearly demonstrated that this population still has barriers to physical exercise. Therefore, we believe that it is essential to know first-hand the experience and experiences of physical exercise, as well as the barriers, facilitators, and reasons for practicing it. In our opinion, it is fundamental to know these types of experiences and realities not only to make better decisions but also to better train the professionals who can work with this population in the context of physical exercise.

The same study suggests: “Considering the aspects mentioned above, the small number of studies on this topic, and the importance of physical activity and sports in the quality of life of the trans population, further research in this area is needed, with the main focus on motivational determinants. The lack of studies regarding the motives of this population to practice is a matter of concern and should be considered in the future.”

The introduction is very long, despite that journal instruction has no limits. The authors reported a lot of information, that makes the readability of this case study very difficult. This introduction could be fit for a full-text article or review. I suggest the authors make it short and focus on the aim of the study.

Response: Dear reviewer, the introduction has been reduced and improved in order to respond to your comment and to make the information presented clearer.

The same thing in the discussion, the authors reported too much information. This discussion form is not acceptable. The authors should shorten the introduction and discussion section accordingly.

Response: Many thanks for the comment and the opportunity to improve the discussion section. The discussion has been changed to make it more objective.

The authors stated that this study is conducted according to the declaration of Helsinki and its subsequent amendments. However, it’s not clear if the author gets the ethical committee approval or not. According to the journal instruction, this type of study should be approved by the IRB.

Response: Thank you for your comment and suggestion. The information regarding the approval of the ethics committee has been added in the manuscript (“The study was conducted in accordance with the Declaration of Helsinki and approved by Ethics Committee of the Polytechnic of Leiria (reference number: CE/IPLEIRIA/20/2022)” ).

The outcome measures used in this study should be reported appropriately in the methods section. The authors should report the validity and reliability of these outcomes.

Response: Thank you very much for your comment. Since this is a case study, the idea of using quantitative instruments was only to complete the information that was collected in the interview. All the instruments used have been translated and validated for the Portuguese population in samples with different characteristics (e.g., young, adult, and elderly; clinical and non-clinical populations), namely:

  • Satisfaction with Life Scale (e.g., Neto, 1993; Antunes et al., 2019);
  • Short Portuguese version of Positive and Negative Affect Schedule (e.g., Galinha et al., 2014; Antunes et al., 2020);
  • Portuguese version of the Hospital Anxiety and Depression Scale (e.g., Pais-Ribeiro, et al., 2007;
  • EUROHIS-QOL 8-item index (Pereira et al., 2011; Antunes et al., 2020; Frontini & Antunes, 2021)

Furthermore, all the instruments used revealed good psychometric properties in their validation processes for the Portuguese population.

Round 2

Reviewer 1 Report

The paper has been significantly revised, it can be published.

Reviewer 2 Report

Good job.

Reviewer 3 Report

I would like to thank the authors for this significant improvement, I have no more comments.